# The combined influence of chronic kidney disease and peripheral artery disease on long-term all-cause and cardio-cerebrovascular disease mortality among middle-aged and elderly individuals: A nationwide cohort study

**Yan-Fang Zhang**[1☯], **Ze-Huang He**[2☯], **Xiao-Feng Zhu**[1], **Guo-Jun Ge**[1], **Xuan Li**[2]*

**1** Department of Nephrology, No. 903 Hospital of PLA Joint Logistic Support Force, Hangzhou Zhejiang, China, **2** Department of Vascular Surgery, Zhejiang Hospital, Hangzhou Zhejiang, China

☯ These authors contributed equally to this work.
* lixuandr79@outlook.com

## Abstract

### Background

The National Health and Nutrition Examination Survey (NHANES) is a nationwide program that evaluates the health and nutritional status of individuals in the United States. Chronic kidney disease (CKD) and peripheral artery disease (PAD) are major contributors to morbidity and mortality, yet their combined impact on mortality outcomes remains uncertain.

### Methods

We analyzed data from NHANES 1999–2004, focusing on individuals aged 40 years and older. CKD was defined by urinary albumin to creatinine ratio (UACR) >30 mg/g or estimated glomerular filtration rate (eGFR) <60 mL/min/1.73 m2, while PAD was determined by an ankle-brachial pressure index (ABI) ≤0.90. Mortality outcomes, including all-cause and cardio-cerebrovascular disease (CCD) mortality, were assessed via linkage to the National Death Index through December 2019.

### Results

Among 7,243 participants, 2,848 all-cause deaths and 921 CCD deaths occurred during a median follow-up of 16.92 years. Participants with concomitant CKD and PAD had higher risks of mortality than those with neither condition or with only one condition. For all-cause mortality, individuals with both CKD and PAD showed the highest risk (adjusted hazard ratio [HR]=3.25, 95% CI: 2.55–4.14, $P < 0.001$). For CCD mortality, the concurrent group likewise had the greatest risk (adjusted HR=4.76, 95% CI: 3.41–6.63, $P < 0.001$).

**Data availability statement:** All relevant data are within the paper and its Supporting Information files.

**Funding:** The author(s) received no specific funding for this work.

**Competing interests:** The authors have no competing interests to declare.

## Conclusion

Coexisting CKD and PAD are associated with substantially elevated risks of all-cause and CCD mortality among middle-aged and older adults. These findings highlight the need for comprehensive, integrated management strategies for individuals with both conditions to mitigate mortality risk.

## Introduction

Approximately 10% of the global population is affected by chronic kidney disease (CKD), which substantially contributes to morbidity and mortality worldwide [1]. CKD is recognized as a leading cause of loss in life expectancy globally and is projected to increase in prominence [2]. Even mild-to-moderate kidney dysfunction is associated with higher risks of cardiovascular disease (CVD) and all-cause mortality [3]. In the United States, an estimated 8.5 million individuals are affected by peripheral artery disease (PAD), and its burden continues to rise [4]. PAD commonly causes exertional leg pain, cramping, and fatigue [4]. Both CKD and PAD are associated with greater morbidity, mortality, and healthcare costs [5,6].

Individuals with CKD have a high prevalence of PAD, typically ranging from 12% to 15% in prior studies [7]. Analyses based on the United States Renal Data System also indicate that PAD is common among patients with CKD, affecting nearly one-quarter of this population [8]. Conversely, patients hospitalized with PAD have a high prevalence of CKD [9]. Evidence further suggests that, in patients with CKD, PAD is associated with substantially increased risks of myocardial infarction, stroke, limb loss, and death [10,11]. O'Hare et al. reported higher mortality among patients with advanced PAD who also had moderate-to-severe CKD [12]. As CKD progresses, the risk of PAD and its complications such as chronic limb-threatening ischemia also increases [13]. Compared with the general population, individuals with both CKD and PAD experience markedly higher rates of adverse outcomes [10]. Given this overlap, it is clinically important to determine whether the coexistence of CKD and PAD confers a greater mortality risk than either condition alone.

CKD and PAD are clinical manifestations of systemic atherosclerosis and are associated with poor survival in the general population [14]. Both are independent risk factors for coronary artery disease [15]. They frequently coexist owing to shared risk factors—including diabetes, hypertension, dyslipidemia, and smoking—and the presence of one condition may exacerbate the other. The ankle–brachial index (ABI) is a simple, noninvasive test used to diagnose PAD, whereas estimated glomerular filtration rate (eGFR) and urinary albumin-to-creatinine ratio (UACR) are established measures of kidney function. Simultaneously assessing ABI-defined PAD together with KDIGO-aligned CKD markers (both eGFR and UACR) provides a pragmatic approach to capture combined vascular–renal burden and to stratify CVD risk in the general population [15]. However, whether the concomitant presence of CKD and PAD confers mortality risk beyond either condition alone—using standardized definitions within a nationally representative sample—remains incompletely characterized.

Prior studies have suggested that individuals with both CKD and PAD experience higher mortality than those with either condition alone; yet many investigations were constrained by modest sample sizes, clinic- or disease-based cohorts, relatively short follow-up, reliance on a single kidney metric (eGFR only), and limited control for complex-survey design and key confounders [10]. Moreover, few studies have examined the potential public health implications of dual screening for CKD and PAD using simple, readily available tests such as eGFR, UACR, and ABI. Given the growing burden of vascular and kidney diseases, identifying individuals with coexisting CKD and PAD in the general population may help target a particularly high-risk subgroup for intensive preventive strategies and multidisciplinary care.

The present study addresses these evidence gaps by leveraging NHANES 1999–2004 with linkage through 2019 (median follow-up 16.9 years), ascertaining PAD by standardized ABI, defining CKD per KDIGO using both eGFR and UACR, and incorporating survey weights, strata, and clusters with comprehensive adjustment for sociodemographic and clinical covariates. Our objective was to evaluate whether the coexistence of CKD and PAD is associated with greater all-cause and cardio-cerebrovascular disease (CCD) mortality than either condition alone, and to highlight its implications for population-level risk stratification and prevention.

## Materials and methods

### Study population

The National Health and Nutrition Examination Survey (NHANES) is a program conducted by the National Center for Health Statistics (NCHS) to assess the health and nutritional status of adults and children in the United States [16]. NHANES utilizes a complex, multistage probability sampling design to select a representative sample of the civilian, non-institutionalized population. This design ensures that the survey results are generalizable to the entire U.S. population. Data collection in NHANES includes interviews, physical examinations, and laboratory tests, providing comprehensive information on a wide range of health-related topics, including demographics, dietary intake, physical activity, and various health conditions. All participants provided written informed consent and study procedures were approved by the National Center for Health Statistics Research Ethics Review Board (Protocol Number: Protocol #98−12).

Participants for this analysis were drawn from the NHANES 1999–2004 cycles and were restricted to individuals aged ≥40 years who completed the lower-extremity disease (LED) examination with valid ankle–brachial index (ABI) measurements. The inclusion and exclusion criteria were defined as follows:

Inclusion criteria: (1) Age ≥ 40 years at baseline; (2) Completed the LED examination with valid ABI data; (3) Availability of kidney function data, including serum creatinine (for eGFR calculation) and/or urinary albumin and creatinine (for UACR calculation). Exclusion criteria: (1) Bilateral lower-limb amputation; (2) Excessive body habitus or other medical conditions precluding accurate ABI measurement; (3) Inability to follow test instructions, acute illness at the time of examination, equipment failure, or missing ABI data due to refusal, early departure, or other logistical issues; (4) Pregnancy at the time of examination; (5) Missing data required for CKD classification. After applying these criteria, 7,255 participants with complete ABI and CKD data were identified. Following linkage to the National Death Index and removal of additional ineligible cases, the final analytic sample consisted of 7,243 participants (**Fig 1**).

### Assessment *of* CKD

The assessment of CKD followed the "Kidney Disease: Improving Global Outcomes (KDIGO) 2021 Guidelines [17]." CKD was defined by a UACR surpassing 30 mg/g or an eGFR falling below 60 mL/min/1.73 m^2. The eGFR was determined via the Chronic Kidney Disease Epidemiology Collaboration equation, incorporating standardized serum creatinine, age, and demographic factors [17].

Laboratory methods utilized for serum creatinine involved the Jaffé reaction with picric acid in an alkaline medium, yielding a yellow-orange creatinine-picric acid complex, measured photometrically. Urinary creatinine analysis employed

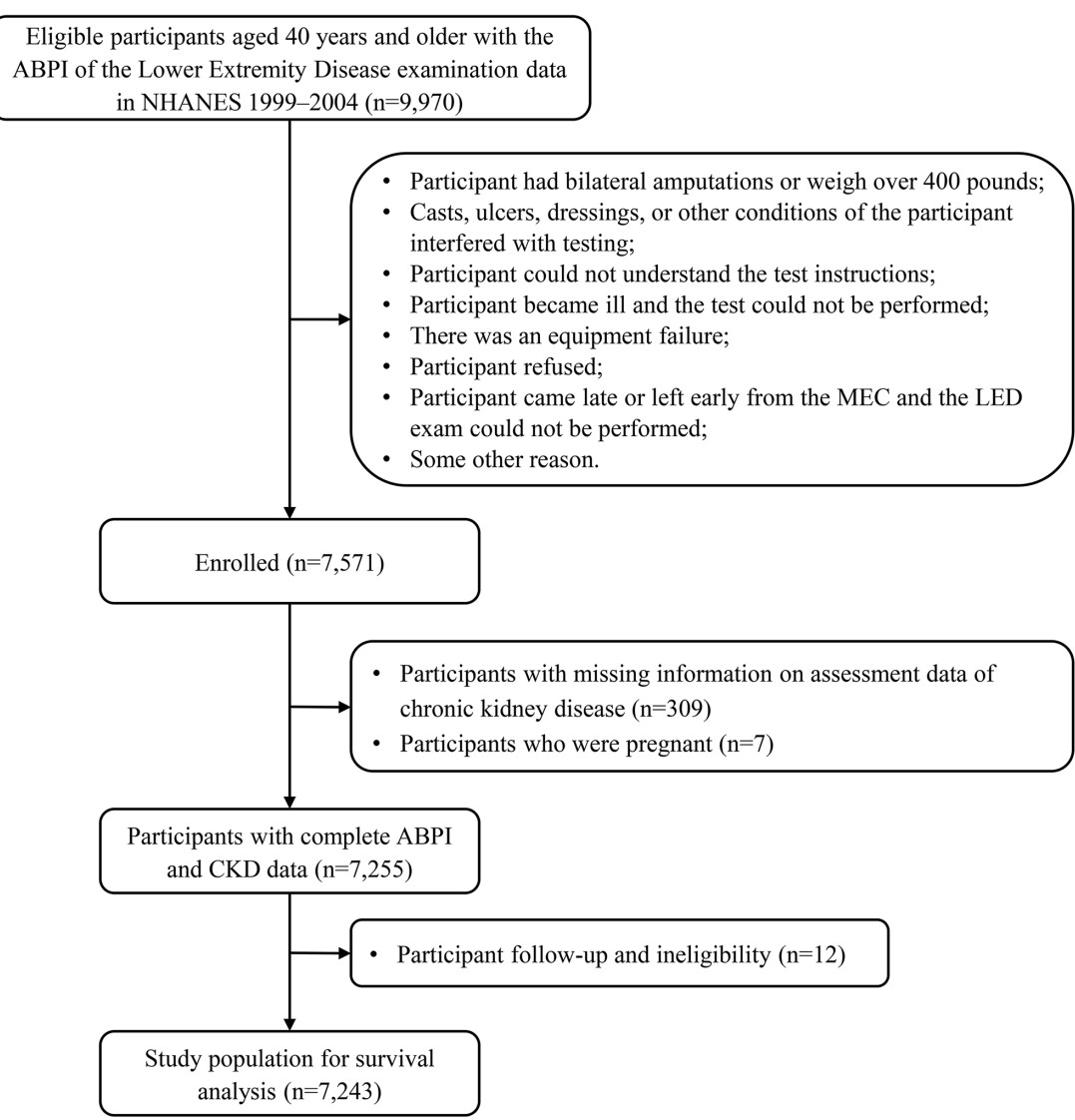

**Fig 1. Flowchart of the study.**

the Jaffé rate reaction, where creatinine reacted with picrate in an alkaline solution to produce a red creatinine-picrate complex, quantified using a CX3 analyzer by measuring the rate difference between 520 nm and 560 nm wavelengths. Urinary albumin assessment employed a solid-phase fluorescent immunoassay, utilizing covalently attached antibodies to human albumin to form complexes with urine specimens, detected with fluorescein-labeled antibodies, and measured using a fluorometer, providing a direct indication of urinary albumin levels.

## Assessment of PAD

Assessment of PAD involves the evaluation of the ABI during the Lower Extremity Disease examination in individuals aged 40 years and older in NHANES 1999–2004. Participants are positioned supine on an examination table, and systolic blood pressure is measured in the right arm and both ankles using an 8-MHz Doppler probe. If conditions such as a rash, open wound, or medical interventions on the right arm preclude accurate measurement, the left arm is utilized. Systolic

blood pressure is measured twice at each site for participants aged 40–59 years and once for those aged 60 years and older. The ABI for each leg is calculated by dividing the ankle pressure by the arm pressure. A diagnosis of PAD is established if the ABI is ≤ 0.90 in either leg [18].

## Assessment *of* mortality

The primary study outcomes encompassed all-cause and CCD mortality [19]. Mortality events were determined by linking NHANES data with death certificate records from the National Death Index (NDI) utilizing a methodology developed by the NCHS. The linked mortality file provides information on follow-up time and the underlying cause of death for NHANES adult participants until December 31, 2019. CCD death was defined as mortality resulting from cerebrovascular or heart diseases, identified through specific International Classification of Diseases, Tenth Revision, codes (I00 to I99).

## Other Covariates

Covariates were selected a priori based on prior literature demonstrating their relevance to both CKD and PAD pathogenesis and their independent associations with mortality outcomes [20,21]. These covariates encompassed demographic characteristics, lifestyle factors, socioeconomic status, and health indicators. Demographic variables comprised age, categorized into two groups: 40–59 years and 60 years or older; sex, categorized as male or female; and race/ethnicity, classified as Non-Hispanic White, Non-Hispanic Black, or Other. Lifestyle factors included living status, categorized as living with partners or living alone; education level, stratified into below high school, high school, or above high school; smoking status, classified as never smoker, former smoker, or current smoker; drinking status, categorized as nondrinker, low-to-moderate drinker, or heavy drinker; and physical activity level, categorized as inactive, insufficiently active, or active. Socioeconomic status was assessed through family poverty income ratio (PIR), classified as ≤1.0, 1.1–3.0, or >3.0. Health indicators comprised body mass index (BMI), categorized as <25.0, 25.0–29.9, or >29.9; adherence to the Healthy Eating Index (HEI), divided into quartiles; and the presence of hypertension, diabetes mellitus, and hyperlipidemia, categorized as yes or no. For comprehensive definitions of family PIR, smoking status, drinking status, physical activity levels, HEI, hypertension, diabetes mellitus, and hyperlipidemia, please refer to the methodology section of the S1 Table.

## Statistical analysis

To ensure the representativeness of our estimates at the national level, we adhered to protocols outlined by the National Center for Health Statistics, incorporating primary sampling units, sample weights, and strata during our data analysis process. Weighted analyses were executed utilizing the "survey" package in R. Continuous variables were presented as medians accompanied by interquartile ranges, with normally distributed variables subjected to Student's t-test. Categorical variables were denoted by numerical values accompanied by percentages. Chi-square testing was employed for assessing unordered categorical variables, while the Kruskal-Wallis H test was utilized for evaluating ordered categorical variables. Any instances of missing data were addressed through imputation utilizing the "mice" package alongside the random forest algorithm.

Survival analysis employed the log-rank test in conjunction with Kaplan-Meier survival curves, whereas COX proportional hazard regression was utilized to explore the relationship between the combined influence of CKD and PAD with mortality among middle-aged and elderly cohorts, with outcomes reported as hazard ratios (HRs) accompanied by their respective 95% confidence intervals (CIs). Assumptions of the Cox proportional hazards model were assessed using Schoenfeld residuals, and no significant violations of the proportional hazards assumption were observed. To evaluate potential multicollinearity in the fully adjusted model, we computed variance inflation factors (VIFs) for all covariates; all VIFs were <5, indicating no concerning multicollinearity (S2 Table). All statistical analyses were conducted using R (version 4.3.2), with statistical significance predetermined at $P < 0.05$.

## Results

### Baseline characteristics *of the* participants

The baseline characteristics of the middle-aged and older participants are presented in **Table 1**, stratified by all-cause mortality status. The total sample size comprised 7243 individuals, among whom 4395 participants were classified as non-mortality cases, while 2848 were classified as mortality cases. The prevalence of CKD was noted in 5.06% of the total cohort, with a higher prevalence observed among individuals who experienced mortality (12.78%) compared to non-mortality cases (1.92%). Similarly, the prevalence of PAD was higher among individuals who experienced mortality (36.97%) compared to non-mortality cases (10.09%).

Regarding demographic characteristics, significant differences were observed across various variables. Notably, mortality cases were more prevalent among individuals aged 60 years or older (71.34%) compared to those aged 40–59 years (28.66%). Additionally, a higher proportion of mortality cases were observed among males (53.08%) compared to females (46.92%). Regarding race/ethnicity, mortality cases showed a higher prevalence among individuals of Non-Hispanic Black ethnicity (9.42%) compared to Non-Hispanic White (80.22%) and other race (10.37%) ethnicities. Furthermore, several lifestyle and health-related factors exhibited significant associations with mortality status. For instance, mortality cases were more prevalent among individuals living alone (61.13%), those with lower education levels (below high school: 31.04%), those with lower family PIR (≤1.0: 13.57%), current smokers (24.48%), nondrinkers (29.13%), physically inactive individuals (35.65%), and those with hypertension (66.38%), diabetes mellitus (23.13%), and hyperlipidemia (83.61%).

**S3** and **S4 Tables** presents baseline characteristics of participants aged 40 years and older from NHANES 1999–2004, stratified by CKD and PAD. Notably, individuals with CKD or PAD exhibit substantially higher rates of all-cause mortality compared to their respective counterparts without these conditions. Moreover, mortality rates attributed to CCD are markedly elevated among those with CKD or PAD relative to those without these conditions.

### Separate association *with* mortality

Among 7,243 participants aged 40 years and older, a total of 2,848 all-cause deaths and 921 CCD deaths were documented during a median follow-up period of 16.92 years (IQR: 15.17–18.67 years). The Kaplan-Meier survival curves depicted in **Fig 2** show a notable trend: participants diagnosed with either CKD or PAD had higher all-cause mortality and CCD mortality than participants without the corresponding two diseases (all P < 0.001). **Table 2** presents the HRs and 95% CIs for mortality associated with CKD or PAD among middle-aged and elderly individuals. For all-cause mortality, individuals with CKD or PAD exhibited a significantly increased risk compared to corresponding individuals without these diseases in all models, respectively. Specifically, in Model 2, the adjusted HR for CKD was 2.00 (95% CI: 1.82–2.20, P < 0.001), indicating a doubling of mortality risk among individuals with CKD compared to those without CKD. Similarly, the adjusted HR for PAD was 1.98 (95% CI: 1.68–2.33, P < 0.001), suggesting a nearly twofold increase in mortality risk among individuals with PAD compared to those without PAD for all-cause mortality. Regarding CCD mortality, individuals with CKD or PAD displayed substantially heightened risks compared to their respective counterparts without these conditions in all models. In Model 2, the adjusted HR for CKD was 2.42 (95% CI: 2.10–2.79, P < 0.001), indicating a 142% increase in mortality risk among individuals with CKD compared to those without CKD. For PAD, the adjusted HR in Model 2 was 2.44 (95% CI: 1.95–3.06, P < 0.001), signifying a 144% increase in mortality risk among individuals with PAD compared to those without PAD for cardio-cerebrovascular disease mortality.

### Combined effects on mortality

The Kaplan-Meier survival curves illustrate that participants concurrently diagnosed with CKD and PAD exhibit higher all-cause (**Fig 3A**) and CCD (**Fig 3B**) mortality rates compared to those without either condition or those with only one of the

**Table 1. Baseline characteristics of the middle-aged and older participants by all-cause mortality in NHANES 1999–2004.**

| Characteristics | Total (n=7243) | All-cause mortality | | P value |
|---|---|---|---|---|
| | | No (n=4395) | Yes (n=2848) | |
| Age, years | | | | <0.001 |
| 40-59 | 3456 (65.11) | 2945 (79.92) | 511 (28.66) | |
| ≥60 | 3787 (34.89) | 1450 (20.08) | 2337 (71.34) | |
| Sex, % | | | | <0.001 |
| Female | 3534 (50.93) | 2294 (52.56) | 1240 (46.92) | |
| Male | 3709 (49.07) | 2101 (47.44) | 1608 (53.08) | |
| Race/ethnicity, % | | | | 0.001 |
| Non-Hispanic White | 3966 (78.29) | 2235 (77.50) | 1731 (80.22) | |
| Non-Hispanic Black | 1253 (8.77) | 774 (8.51) | 479 (9.42) | |
| Other race | 2024 (12.94) | 1386 (13.99) | 638 (10.37) | |
| Living status, % | | | | <0.001 |
| With partners | 2458 (29.24) | 1257 (25.33) | 1201 (38.87) | |
| Alone | 4785 (70.76) | 3138 (74.67) | 1647 (61.13) | |
| Education level, % | | | | <0.001 |
| Below high school | 2443 (19.86) | 1261 (15.31) | 1182 (31.04) | |
| High school | 1689 (26.05) | 995 (24.98) | 694 (28.68) | |
| Above high school | 3111 (54.09) | 2139 (59.70) | 972 (40.28) | |
| Family PIR, % | | | | <0.001 |
| ≤1.0 | 1108 (10.21) | 591 (8.84) | 517 (13.57) | |
| 1.1–3.0 | 3040 (33.30) | 1600 (27.81) | 1440 (46.82) | |
| >3.0 | 3095 (56.49) | 2204 (63.35) | 891 (39.61) | |
| Smoking status, % | | | | <0.001 |
| Never smoker | 3354 (46.07) | 2235 (49.75) | 1119 (37.02) | |
| Former smoker | 2487 (33.17) | 1342 (31.01) | 1145 (38.50) | |
| Current smoker | 1402 (20.76) | 818 (19.25) | 584 (24.48) | |
| Drinking status, % | | | | <0.001 |
| Nondrinker | 1865 (22.51) | 1014 (19.82) | 851 (29.13) | |
| Low-to-moderate drinker | 4746 (67.33) | 3020 (70.71) | 1726 (59.01) | |
| Heavy drinker | 632 (10.16) | 361 (9.47) | 271 (11.86) | |
| Body mass index, % | | | | 0.061 |
| <25.0 kg/m$^2$ | 1971 (28.84) | 1109 (27.86) | 862 (31.27) | |
| 25.0-29.9 kg/m$^2$ | 2836 (38.17) | 1731 (38.77) | 1105 (36.70) | |
| >29.9 kg/m$^2$ | 2436 (32.98) | 1555 (33.37) | 881 (32.02) | |
| Physical activity, % | | | | <0.001 |
| Inactive | 2276 (24.16) | 1147 (19.50) | 1129 (35.65) | |
| Insufficiently active | 3406 (54.49) | 2286 (59.20) | 1120 (42.90) | |
| Active | 1561 (21.35) | 962 (21.30) | 599 (21.45) | |
| HEI | 51.14 (42.14,60.43) | 50.88 (42.04,60.27) | 51.56 (42.49,60.89) | 0.158 |
| Hypertension, % | | | | <0.001 |
| No | 3299 (52.86) | 2435 (60.68) | 864 (33.62) | |
| Yes | 3944 (47.14) | 1960 (39.32) | 1984 (66.38) | |
| Diabetes mellitus, % | | | | <0.001 |
| No | 5935 (86.83) | 3820 (90.87) | 2115 (76.87) | |
| Yes | 1308 (13.17) | 575 (9.13) | 733 (23.13) | |

*(Continued)*

**Table 1.** (Continued)

| Characteristics | Total (n=7243) | All-cause mortality | | P value |
| --- | --- | --- | --- | --- |
| | | No (n=4395) | Yes (n=2848) | |
| Hyperlipidemia, % | | | | < 0.001 |
| No | 1444 (19.91) | 907 (21.33) | 537 (16.39) | |
| Yes | 5799 (80.09) | 3488 (78.67) | 2311 (83.61) | |
| CKD, % | | | | <0.001 |
| No | 6682 (94.94) | 4273 (98.08) | 2409 (87.22) | |
| Yes | 561 (5.06) | 122 (1.92) | 439 (12.78) | |
| PAD, % | | | | <0.001 |
| No | 5488 (82.14) | 3838 (89.91) | 1650 (63.03) | |
| Yes | 1755 (17.86) | 557 (10.09) | 1198 (36.97) | |
| Follow-up time, years | 16.92 (15.17,18.67) | 17.83 (16.50,19.17) | 10.42 (6.17,14.50) | <0.001 |

Abbreviations: PIR, poverty income ratio; HEI, Healthy Eating Index; CKD, chronic kidney disease; PAD, peripheral artery disease.

Continuous variables are presented as medians [interquartile ranges]. Categorical variables are presented as numbers (percentages). Sampling weights were applied for calculation of demographic descriptive statistics; N reflect the study sample while percentages reflect the survey-weighted data.

diseases (all *P*<0.001). Table 3 presents the HRs and 95% CIs for mortality based on the combined influence of CKD and PAD among middle-aged and elderly individuals. For all-cause mortality, individuals with CKD alone exhibited a significantly increased risk compared to those without CKD and PAD, with a Model 2 adjusted HR of 1.99 (95% CI: 1.81–2.19, P<0.001). Similarly, individuals with PAD alone also demonstrated a significantly elevated risk of mortality compared to those without CKD and PAD, with a Model 2 adjusted HR of 2.05 (95% CI: 1.64–2.55, P<0.001). Notably, individuals with both CKD and PAD showed the highest risk of mortality, with a Model 2 adjusted HR of 3.25 (95% CI: 2.55–4.14, P<0.001). For CCD mortality, similar patterns were observed. Individuals with CKD alone had a significantly increased risk compared to those without CKD and PAD, with a Model 2 adjusted HR of 2.40 (95% CI: 2.05–2.81, P<0.001). Likewise, individuals with PAD alone exhibited a significantly elevated risk compared to those without CKD and PAD, with a Model 2 adjusted HR of 2.51 (95% CI: 1.83–3.43, P<0.001). Individuals with both CKD and PAD demonstrated the highest risk of CCD mortality, with a Model 2 adjusted HR of 4.76 (95% CI: 3.41–6.63, P<0.001).

### Sensitivity analysis

The sensitivity analyses conducted to explore the influence of excluding participants with specific medical histories at baseline on the mortality associated with the combined presence of CKD and PAD. Across all analyses, exclusion criteria for participants who died within two years of follow-up (S5 Table), had a history of CVD at baseline (S6 Table), or had a history of cancer at baseline (S7 Table) were applied. The results consistently showed that individuals with both CKD and PAD exhibited significantly elevated risks of all-cause and CCD mortality compared to those without either condition or those with only one of the diseases, even after adjusting for various demographic and clinical factors.

### Discussion

In our study, we discussed the combined influence of CKD and PAD on long-term all-cause and CCD mortality among 7,243 middle-aged and elderly individuals based on data with a median follow-up of 16.92 years from NHANES 1999–2004. The results suggested that patients with both CKD and PAD had an increased risk of all-cause and CCD mortality among middle-aged and elderly individuals compared to those without these diseases or those with one disease alone. Even after accounting for traditional CVD risk factors, the elevated risk of mortality remained.

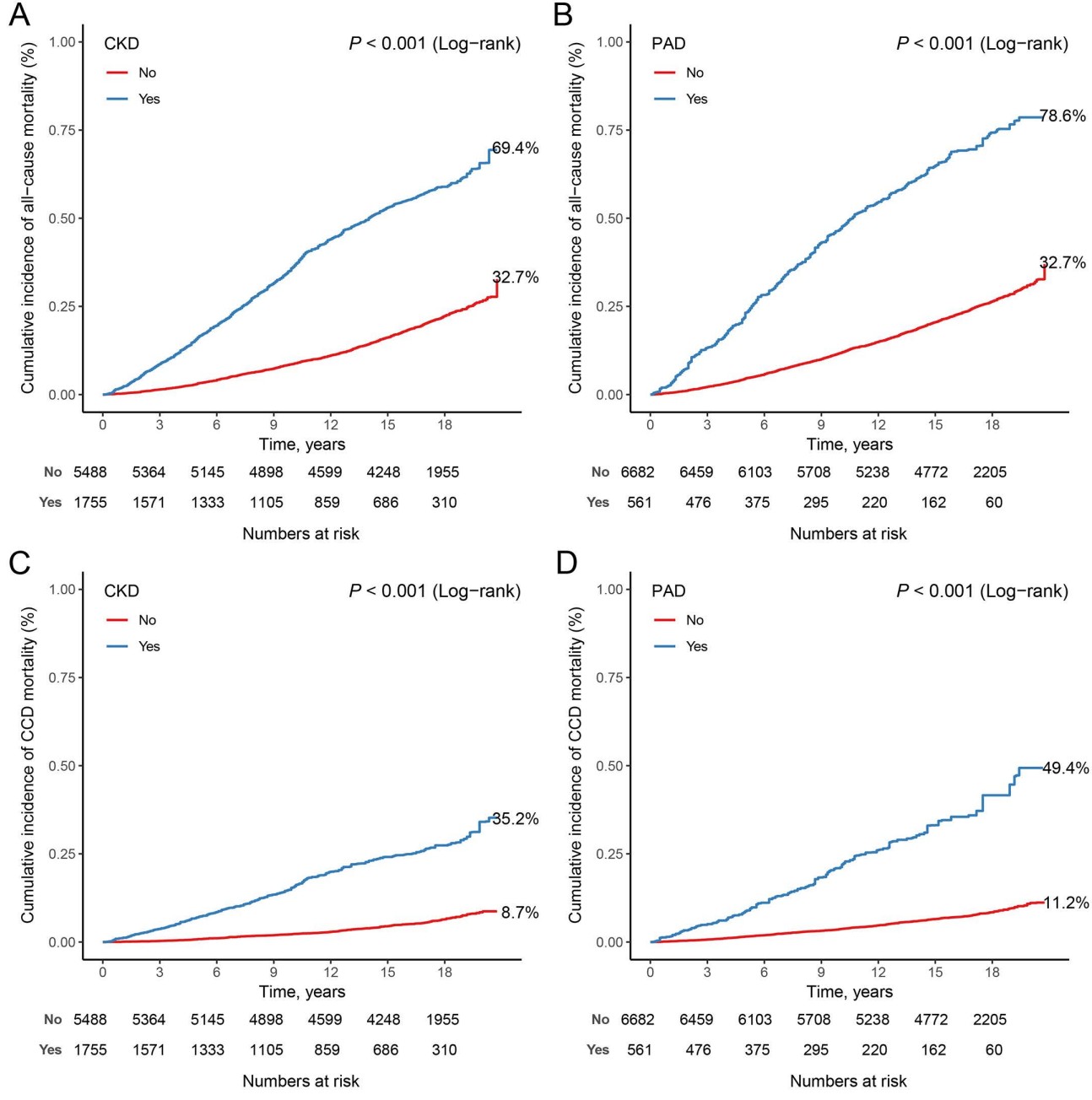

**Fig 2. Kaplan-Meier survival curves for participants diagnosed with either CKD or PAD and mortality among middle-aged and elderly individuals.** (A) CKD and all-cause mortality; (B) PAD and all-cause mortality; (C) CKD and CCD mortality; (D) PAD and CCD mortality.

CKD and PAD frequently coexist and may influence each other's progression [5,6]. Declining kidney function contributes to the development and progression of PAD: as the GFR decreases, the risk of PAD increases in both the general population and among individuals with CKD [22]. A recent study examined the ankle–brachial index–toe–brachial index (ABI–TBI), a composite measure used to assess atherosclerosis and arterial stiffness, in relation to all-cause mortality among patients with CKD, and found that a higher ABI–TBI was associated with greater mortality risk [23]. Although CKD and PAD share common

**Table 2. HRs (95% CIs) of mortality according to CKD or PAD among middle-aged and elderly individuals in NHANES 1999–2004.**

| | Crude | | Model 1 | | Model 2 | |
|---|---|---|---|---|---|---|
| | HR (95% CI) | *P* value | HR (95% CI) | *P* value | HR (95% CI) | *P* value |
| **All-cause mortality** | | | | | | |
| CKD | | | | | | |
| No | 1 [Reference] | | 1 [Reference] | | 1 [Reference] | |
| Yes | 3.86 (3.50-4.26) | <0.001 | 2.38 (2.18-2.60) | <0.001 | 2.00 (1.82-2.20) | <0.001 |
| PAD | | | | | | |
| No | 1 [Reference] | | 1 [Reference] | | 1 [Reference] | |
| Yes | 4.57 (3.92-5.34) | <0.001 | 2.71 (2.32-3.17) | <0.001 | 1.98 (1.68-2.33) | <0.001 |
| **CCD Mortality** | | | | | | |
| CKD | | | | | | |
| No | 1 [Reference] | | 1 [Reference] | | 1 [Reference] | |
| Yes | 5.29 (4.58-6.11) | <0.001 | 3.15 (2.73-3.64) | <0.001 | 2.42 (2.10-2.79) | <0.001 |
| PAD | | | | | | |
| No | 1 [Reference] | | 1 [Reference] | | 1 [Reference] | |
| Yes | 5.97 (4.87-7.34) | <0.001 | 3.38 (2.74-4.17) | <0.001 | 2.44 (1.95-3.06) | <0.001 |

Model 1 was adjusted for age (40–59, or ≥60), sex (male or female), and race/ethnicity (Non-Hispanic White, Non-Hispanic Black or Other); Model 2 was adjusted as model 1 plus living status (with partners, or alone), education level (below high school, high school, or above high school), family PIR (≤1.0, 1.1–3.0, or >3.0), smoking status (never smoker, former smoker, or current smoker), drinking status (nondrinker, low-to-moderate drinker, or heavy drinker), BMI (<25.0, 25.0–29.9, or >29.9), physical activity (inactive, insufficiently active, or active), HEI (in quartiles), hypertension (yes or no), diabetes mellitus (yes or no), and hyperlipidemia (yes or no).

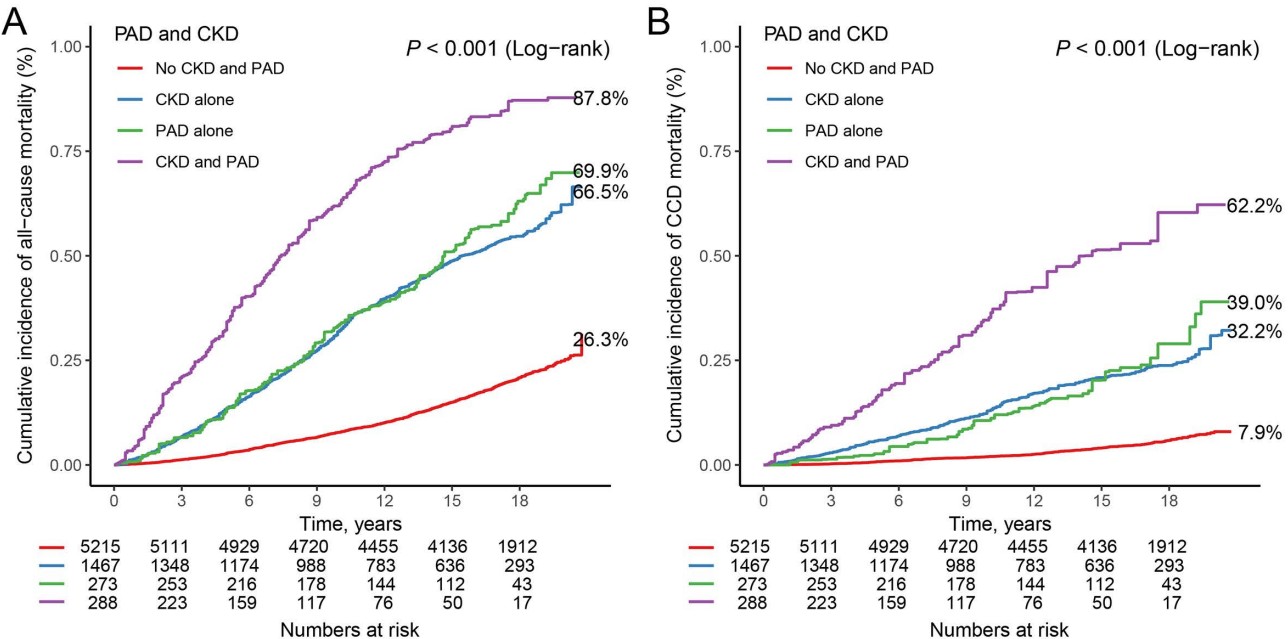

**Fig 3. Kaplan-Meier survival curves for participants diagnosed with CKD and PAD and all-cause (A) and CCD (B) mortality among middle-aged and elderly individuals.**

**Table 3.** HRs (95% CIs) of mortality according to the combined influence of CKD and PAD among middle-aged and elderly individuals in NHANES 1999–2004.

| | Crude | | Model 1 | | Model 2 | | P-int |
|---|---|---|---|---|---|---|---|
| | HR (95% CI) | P value | HR (95% CI) | P value | HR (95% CI) | P value | |
| **All-cause mortality** | | | | | | | 0.178 |
| No CKD and PAD | 1 [Reference] | | 1 [Reference] | | 1 [Reference] | | |
| CKD alone | 3.68 (3.35-4.04) | <0.001 | 2.34 (2.15-2.54) | <0.001 | 1.99 (1.81-2.19) | <0.001 | |
| PAD alone | 4.24 (3.35-5.36) | <0.001 | 2.75 (2.24-3.37) | <0.001 | 2.05 (1.64-2.55) | <0.001 | |
| CKD and PAD | 10.04 (7.79-12.93) | <0.001 | 4.79 (3.85-5.96) | <0.001 | 3.25 (2.55-4.14) | <0.001 | |
| **CCD Mortality** | | | | | | | 0.354 |
| No CKD and PAD | 1 [Reference] | | 1 [Reference] | | 1 [Reference] | | |
| CKD alone | 4.97 (4.20-5.88) | <0.001 | 3.07 (2.60-3.62) | <0.001 | 2.40 (2.05-2.81) | <0.001 | |
| PAD alone | 5.36 (3.74-7.69) | <0.001 | 3.36 (2.38-4.75) | <0.001 | 2.51 (1.83-3.43) | <0.001 | |
| CKD and PAD | 16.49 (12.18-22.34) | <0.001 | 7.54 (5.67-10.03) | <0.001 | 4.76 (3.41-6.63) | <0.001 | |

Model 1 was adjusted for age (40–59, or ≥60), sex (male or female), and race/ethnicity (Non-Hispanic White, Non-Hispanic Black or Other); Model 2 was adjusted as model 1 plus living status (with partners, or alone), education level (below high school, high school, or above high school), family PIR (≤1.0, 1.1–3.0, or >3.0), smoking status (never smoker, former smoker, or current smoker), drinking status (nondrinker, low-to-moderate drinker, or heavy drinker), BMI (<25.0, 25.0–29.9, or >29.9), physical activity (inactive, insufficiently active, or active), HEI (in quartiles), hypertension (yes or no), diabetes mellitus (yes or no), and hyperlipidemia (yes or no). *P-int, P* for interaction.

risk factors (e.g., diabetes, hypertension, smoking), emerging evidence suggests that CKD itself may be an independent risk factor for PAD [24,25]. CKD is characterized by chronic inflammation; therefore, in individuals with CKD, the interplay among inflammation, oxidative stress, and other pathophysiological processes likely contributes to the pathogenesis and progression of PAD [24]. CKD is also associated with dysregulation of angiogenic factors—including vascular endothelial growth factor (VEGF), endostatin, and circulating endothelial cells—which may impair angiogenesis and vascular function [26].

CKD is further associated with microvascular rarefaction [27], which, in the context of PAD, can exacerbate tissue ischemia and impair perfusion. The accumulation of uremic toxins due to impaired kidney function has been implicated in CVD complications, including atherosclerosis and PAD [28]. In addition, experimental and translational work evaluating aryl hydrocarbon receptor (AHR) activation in skeletal muscle—using mouse models and human data—suggests that genetic deletion of AHR attenuates ischemic limb progression and reduces the risk of amputation or death [29]. A more comprehensive understanding of these mechanisms may inform the development of targeted therapies to reduce CVD complications in CKD and improve clinical outcomes.

Patients with PAD were also independently associated with the development of CKD [30]. It has also been reported to predict increases in serum creatinine levels [31]. Other evidence also indicated that PAD played a crucial role in the renal outcomes and mortality risk in patients. Noppawit et al. conducted a multicenter prospective cohort study to explore the relationships between ABI and renal outcomes and all-cause mortality in 5543 Thailand patients with high CVD risk [32]. PAD and CKD likely share overlapping pathogenic pathways such as endothelial dysfunction (characterized by reduced nitric oxide bioavailability) [33–35]. The association between moderate-to-severe atherosclerosis and renal structural changes—including sclerotic glomeruli and increased intrarenal arterial wall area—reported by Kasiske et al. underscores the tight linkage between CVD and renal pathology [36]. In addition, Zamami et al. observed that individuals with a high-normal ABI (1.23) had greater intimal thickness of small renal arteries and lower eGFR, further supporting vascular–renal coupling [37]. Collectively, these data support early detection and treatment of PAD to mitigate CKD progression.

When PAD and CKD coexist, clinical outcomes are worse than with either condition alone [38]. The coexistence is associated with higher amputation rates, likely reflecting impaired limb perfusion due to PAD combined with diminished wound healing and heightened infection susceptibility in CKD [38]. Liew et al. reported a 1.5-fold higher mortality over six

years in individuals with both conditions compared with those with only one [10], and another study similarly found higher mortality over four years versus CKD alone [39]. Florian et al. showed that CKD was associated with a threefold increase in both in-hospital and long-term mortality among patients with PAD, with additive detrimental effects on health outcomes and healthcare costs [38]. In our study, which included a substantially larger sample and longer follow-up than prior studies, these findings were reaffirmed. For all-cause mortality, individuals with both CKD and PAD had the highest risk, with an adjusted HR of 3.25 over a median follow-up of 16.92 years.

Moreover, numerous studies have demonstrated that CKD and PAD are independently associated with CVD ischemic outcomes [40,41]. Both conditions exert substantial adverse effects on cardiovascular health, and their coexistence may amplify risk, a phenomenon often described as "double jeopardy." Prior studies reported long-term cardiovascular event rates approaching 28.3% in such high-risk populations, indicating that nearly one-third of affected individuals experience events over time [42]. These investigations further suggest that PAD and CKD each independently contribute to CVD risk [42]. Consistent with this, our findings showed that for CCD mortality, individuals with both conditions had the greatest risk, with an adjusted HR of 4.76. These results underscore the significant prognostic impact of concurrent CKD and PAD on both overall and CCD-specific mortality.

The main strength of this study is the use of simple, validated, and clinically accessible measures to ascertain both CKD and PAD. CKD was defined using eGFR and UACR, and PAD was assessed with the ankle–brachial index (ABI), a sensitive and specific diagnostic tool. These standardized criteria facilitate early identification of at-risk individuals and enable timely interventions to slow disease progression and prevent complications. Incorporating both ABI and CKD assessments into routine clinical practice may enhance risk stratification and support individualized management strategies for patients with multimorbidity. Besides, our study also benefited from a large, nationally representative sample and long-term mortality follow-up, allowing for robust estimates of risk. CKD and PAD share several modifiable risk factors such as hypertension, diabetes, smoking, and psychological factors [43,44]. This highlights the importance of integrated risk factor modification to reduce the cumulative burden of both conditions. Our findings suggest that the coexistence of CKD and PAD should be recognized as a high-risk phenotype that warrants more intensive monitoring and multidisciplinary care approaches.

Some limitations in the present study should be in consideration. Firstly, owing to the observational cohort design with single baseline measurements, we cannot establish causality between CKD, PAD, their combination, and subsequent mortality. Accordingly, our findings should be interpreted as associations rather than causal effects. Moreover, the absence of lower-limb angiography which is considered the gold standard for diagnosing [45], the use of ABI alone may lead to misclassification of PAD in individuals with arterial stiffness, particularly among those with diabetes or CKD, due to falsely elevated ABI values. Third, despite comprehensive adjustment, residual and unmeasured confounding cannot be excluded—particularly limited information on medication use (including antiplatelet/anticoagulant therapy, statins, or renin–angiotensin system), inflammatory markers, treatment intensity, and adherence to CKD/PAD management and lifestyle interventions. Fourth, reliance on single baseline measurements of eGFR, UACR, and ABI may not capture longitudinal changes. Finally, the cohort primarily comprised middle-aged and older adults, which may limit generalizability to younger populations.

## Conclusion

In this nationally representative cohort of middle-aged and older adults, the coexistence of CKD and PAD was associated with incrementally higher risks of both all-cause and CCD mortality than either condition alone. These associations persisted after multivariable adjustment, consistent with an additive, rather than synergistic, pattern of risk. Collectively, the findings underscore the need for targeted prevention and coordinated, integrated management, including routine incorporation of ABI and kidney assessments, optimized risk-factor management, and multidisciplinary care for individuals with coexisting CKD and PAD.

## Supporting information

**S1 Table. Definitions and classification criteria for covariates included in the analysis.**
(DOCX)

**S2 Table. Variance inflation factor (VIF) values for covariates included in the multivariable Cox regression model.**
(DOCX)

**S3 Table. Baseline characteristics of the middle-aged and older participants by CKD in NHANES 1999–2004.**
(DOCX)

**S4 Table. Baseline characteristics of the middle-aged and older participants by PAD in NHANES 1999–2004.**
(DOCX)

**S5 Table. HRs (95% CIs) of mortality according to the combined influence of CKD and PAD after excluding participants who died within two years of follow-up among middle-aged and elderly individuals in NHANES 1999–2004.**
(DOCX)

**S6 Table. HRs (95% CIs) of mortality according to the combined influence of CKD and PAD after excluding participants who had CVD history at baseline among middle-aged and elderly individuals in NHANES 1999–2004.**
(DOCX)

**S7 Table. HRs (95% CIs) of mortality according to the combined influence of CKD and PAD after excluding participants who had cancer history at baseline among middle-aged and elderly individuals in NHANES 1999–2004.**
(DOCX)

## Acknowledgments

We appreciate the people who contributed to the NHANES data we studied.

## Author contributions

**Data curation:** Yan-Fang Zhang.

**Funding acquisition:** Yan-Fang Zhang.

**Project administration:** xuan li.

**Supervision:** Guo-Jun Ge, xuan li.

**Validation:** Ze-Huang He.

**Writing – original draft:** Yan-Fang Zhang, Ze-Huang He, Xiao-Feng Zhu, Guo-Jun Ge.

**Writing – review & editing:** Yan-Fang Zhang.

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
