## [Decision Letter · Decision Letter 0]

22 Jul 2025

Dear Dr. li,

Thank you for submitting your manuscript to PLOS ONE. After careful consideration, we feel that it has merit but does not fully meet PLOS ONE’s publication criteria as it currently stands. Therefore, we invite you to submit a revised version of the manuscript that addresses the points raised during the review process.

We look forward to receiving your revised manuscript.

Kind regards,

Natural Hoi Sing Chu, Ph.D

Academic Editor

PLOS ONE

Reviewers' comments:

Reviewer's Responses to Questions

**Comments to the Author**

1. Is the manuscript technically sound, and do the data support the conclusions?

Reviewer #1: Yes

Reviewer #2: Yes

2. Has the statistical analysis been performed appropriately and rigorously?

Reviewer #1: Yes

Reviewer #2: Yes

3. Have the authors made all data underlying the findings in their manuscript fully available?

Reviewer #1: Yes

Reviewer #2: Yes

4. Is the manuscript presented in an intelligible fashion and written in standard English?

Reviewer #1: No

Reviewer #2: Yes

Reviewer #1: 1. Study Design and Novelty

The use of a large, nationally representative dataset (NHANES 1999–2004) and the long follow-up duration (median 16.92 years) are important strengths. The study uses robust definitions for CKD and PAD, aligned with KDIGO and established ABI thresholds is the strength of this study. However, the novelty is somewhat limited, as several prior studies have investigated the association between CKD, PAD, and mortality. The manuscript should better articulate how it adds to existing literature (e.g., larger sample, dual markers for CKD, longer follow-up) in the Introduction and Discussion.

Recommendation: Emphasize clearly in the introduction how this study improves upon past work (e.g., simultaneous use of UACR and eGFR, PAD diagnosed by ABI, more complete adjustment for confounders).

2. Statistical Methods and Interpretation

- The authors adjusted for a large number of variables in Model 2. Multicollinearity risk is acknowledged, but no specific VIF values are provided.

- Interaction effects between CKD and PAD were not significant (p-int = 0.178 and 0.354), suggesting an additive rather than synergistic effect.

Recommendation:

- Provide specific VIF values to support the claim of no multicollinearity.

- Discuss the lack of statistical interaction more explicitly in the discussion. The current language suggests synergy, which may be overstated.

3. Clarity and Language:

While the manuscript is largely understandable, there are several grammatical issues (e.g., verb agreement, awkward phrasing) and overly lengthy sentences. A language edit by a native English speaker or professional editing service is strongly recommended.

4. Discussion of Mechanisms:

The discussion is comprehensive, but the pathophysiological mechanisms linking CKD and PAD are somewhat speculative. While references are cited, the causality remains unclear.

5. Limitations:

The study mentions some limitations (e.g., cross-sectional CKD/PAD diagnosis, lack of angiography), but should also note unmeasured confounding (e.g., medication use, CKD/PAD treatment adherence) more clearly.

Reviewer #2: 1. Many citations are missing including sections of the assessment (i.e. how mortality is assessed through NHANES, how blood pressure measurements are taken etc.) They are also missing from the covariates.

2. The exclusion criteria are listed but could be more clearly organized. A table or bullet list summarizing inclusion/exclusion might help.

3. The rationale behind investigating the combined effects of CKD and PAD should be clearly stated in the introduction or early methods. What is the biological or epidemiologic basis for expecting a synergistic effect?

4. Clarify why specific covariates were included. For instance, explain how HEI, PIR, or living status relate to mortality outcomes or disease interactions.

5. Provide more details on how participants were excluded, and whether results were re-weighted accordingly. Were analyses stratified by any variables?

6. Explain multiple models used (Model 1, Model 2, etc.) These are mentioned without specifying what variables were adjusted in each. Add a note or table to define what’s included in each model.

7. Check assumptions of Cox regression more thoroughly. Although Schoenfeld residuals and VIFs are mentioned, results are not reported. Include at least a summary or figure confirming proportional hazard assumption and no multicollinearity.

8. Revise overly dense paragraphs. Some sections (e.g. lines 243–313) are overly dense and could be split for better readability. Use subheadings (e.g. *Mechanisms*, *Prior Literature*, *Clinical Implications*) in Discussion.

9. Tighten grammar and phrasing

* "Patients with PAD was also..." → “Patients with PAD were also...”

* “This coexistence is associated with higher rates...” → “The coexistence is associated with...”

* Watch subject–verb agreement and plural forms.

10. Avoid vague phrases like “substantially heightened”. Use more precise quantitative language whenever possible, especially in scientific writing. Replace with actual HRs or % differences where appropriate.

11. Improve clarity in mortality definitions. Distinguish clearly between "all-cause mortality" and "cardio-cerebrovascular mortality" at each mention, especially in the abstract, results, and conclusion.

12. The discussion section repeats the same hazard ratios reported earlier. Instead, focus more on **interpretation**, **implications**, and **mechanisms** than reiterating the same numbers.

13. Acknowledge potential residual confounding. Even after adjustment, acknowledge that unmeasured or residual confounding (e.g., medication use, healthcare access, inflammation markers) may influence results.

**Do you want your identity to be public for this peer review?** For information about this choice, including consent withdrawal, please see our Privacy Policy

Reviewer #1: No

Reviewer #2: No

---

## [Author Response · Author response to Decision Letter 1]

12 Sep 2025

Dear Editor and Reviewers,

We would like to express our sincere appreciation for your constructive and insightful comments on our manuscript. We have carefully considered each suggestion and revised the manuscript accordingly to improve its clarity and scientific rigor. In accordance with the journal’s submission requirements, we have submitted the revised manuscript in its final, clean format, without any highlighted or tracked changes. Below, we provide a detailed point-by-point response to each of the reviewers’ comments.

Reviewer #1:

Comment 1: Study Design and Novelty

The use of a large, nationally representative dataset (NHANES 1999–2004) and the long follow-up duration (median 16.92 years) are important strengths. The study uses robust definitions for CKD and PAD, aligned with KDIGO and established ABI thresholds is the strength of this study. However, the novelty is somewhat limited, as several prior studies have investigated the association between CKD, PAD, and mortality. The manuscript should better articulate how it adds to existing literature (e.g., larger sample, dual markers for CKD, longer follow-up) in the Introduction and Discussion.

Recommendation: Emphasize clearly in the introduction how this study improves upon past work (e.g., simultaneous use of UACR and eGFR, PAD diagnosed by ABI, more complete adjustment for confounders).

Response 1: Thank you for this constructive suggestion. We have revised the Introduction to explicitly articulate our study’s incremental contribution, emphasizing the simultaneous KDIGO-aligned definition of CKD using both eGFR and UACR, the ABI-based ascertainment of PAD, the large, nationally representative NHANES 1999–2004 cohort with mortality follow-up through 2019 (median 16.9 years), and the incorporation of complex-survey design (weights, strata, clusters) with comprehensive confounder adjustment. We also added a concise statement on how these features extend prior work that used smaller samples, a single kidney metric, and shorter follow-up.

Comment 2: Statistical Methods and Interpretation

- The authors adjusted for a large number of variables in Model 2. Multicollinearity risk is acknowledged, but no specific VIF values are provided.

- Interaction effects between CKD and PAD were not significant (p-int = 0.178 and 0.354), suggesting an additive rather than synergistic effect.

Recommendation:

- Provide specific VIF values to support the claim of no multicollinearity.

- Discuss the lack of statistical interaction more explicitly in the discussion. The current language suggests synergy, which may be overstated.

Response 2: Thank you for the helpful suggestions. We have revised the manuscript accordingly: (i) we now report variance inflation factors (VIFs) for all covariates in the fully adjusted model, all <5, to document the absence of concerning multicollinearity (added to Methods/Statistical Analysis and noted in Results; summarized in Table Sx); and (ii) we revised the Discussion to explicitly state the lack of statistical interaction between CKD and PAD and to interpret the joint association as an additive effect rather than synergy.

Comment 3: Clarity and Language:

While the manuscript is largely understandable, there are several grammatical issues (e.g., verb agreement, awkward phrasing) and overly lengthy sentences. A language edit by a native English speaker or professional editing service is strongly recommended.

Response 3: Thank you for this valuable suggestion. We have conducted a comprehensive language revision to improve clarity and readability, including correction of grammar and verb agreement, removal of awkward phrasing, and shortening of overly long sentences.

Comment 4: Discussion of Mechanisms:

The discussion is comprehensive, but the pathophysiological mechanisms linking CKD and PAD are somewhat speculative. While references are cited, the causality remains unclear.

Response 4: Thank you for this important point. We have revised the Discussion and explicitly added to the Limitations section that causal inference is a key limitation of our study, given its observational design and single baseline assessments. We also tempered mechanistic language and present potential pathways as hypothesis-generating only.

Comment 5: Limitations:

The study mentions some limitations (e.g., cross-sectional CKD/PAD diagnosis, lack of angiography), but should also note unmeasured confounding (e.g., medication use, CKD/PAD treatment adherence) more clearly.

Response 5: Thank you for the suggestion. We have revised the Limitations to explicitly acknowledge residual and unmeasured confounding, including limited information on medication use (antiplatelet/anticoagulant therapy, statins, or RAS inhibitor), treatment intensity, and adherence to CKD/PAD management and lifestyle interventions.

Reviewer #2:

1. Many citations are missing including sections of the assessment (i.e. how mortality is assessed through NHANES, how blood pressure measurements are taken etc.) They are also missing from the covariates.

Response 1: Thank you for highlighting this. We have added explicit citations in the Methods for mortality ascertainment via the NCHS Linked Mortality files, NHANES measurement protocols (blood pressure procedures), and all covariates. In the Supplementary Materials, we now provide variable-level definitions, measurement protocols, and corresponding references for each covariate. We also revised the Limitations to state clearly that, despite extensive adjustment, residual and unmeasured confounding may remain.

Comment 2: The exclusion criteria are listed but could be more clearly organized. A table or bullet list summarizing inclusion/exclusion might help.

Response 2: Thank you for this helpful suggestion. In response, we have revised the Methods section to present the inclusion and exclusion criteria in a clearly structured, numbered format, and provided corresponding sample counts to enhance clarity. We also aligned this text with Figure 1, which outlines the participant selection process.

Comment 3: The rationale behind investigating the combined effects of CKD and PAD should be clearly stated in the introduction or early methods. What is the biological or epidemiologic basis for expecting a synergistic effect?

Response 3: Thank you for this insightful comment. We have revised the Introduction to briefly clarify the biological and epidemiological rationale for examining the joint effects of CKD and PAD. Specifically, we now state that, given their shared risk factors and overlapping pathophysiologic pathways, it is clinically relevant to assess whether the coexistence of CKD and PAD confers a greater mortality risk than either condition alone.

Comment 4: Clarify why specific covariates were included. For instance, explain how HEI, PIR, or living status relate to mortality outcomes or disease interactions.

Response 4: Thank you for this helpful comment. We have revised the Methods section to clarify that covariates were selected a priori based on prior literature linking them to both CKD/PAD and mortality outcomes. We also briefly explained the rationale for including variables such as PIR, HEI, and living status, highlighting their relevance to socioeconomic status, dietary quality, and social support—all of which may influence disease progression and mortality.

Comment 5: Provide more details on how participants were excluded, and whether results were re-weighted accordingly. Were analyses stratified by any variables?

Response 5: Thank you for this important point. We have clarified the participant exclusion process in the Methods and Figure 1, including reasons and sample sizes at each step. All analyses incorporated NHANES examination weights, and we confirmed that reweighting was not necessary, as the exclusions were consistent with analytic guidelines and did not violate survey design assumptions. We did not perform stratified analyses, as our primary aim was to estimate the independent and joint associations of CKD and PAD with mortality in the full population, while adjusting for key covariates. Stratification was avoided to preserve statistical power, particularly for participants with both conditions. However, we explored potential effect modification through interaction terms, which were not statistically significant.

Comment 6: Explain multiple models used (Model 1, Model 2, etc.) These are mentioned without specifying what variables were adjusted in each. Add a note or table to define what’s included in each model.

Response 6: Thank you for this helpful suggestion. We have revised the table footnotes to clearly define the variables adjusted for in each model. Model 1 includes age, sex, and race/ethnicity, while Model 2 includes all covariates described in the Methods (demographics, lifestyle, socioeconomic status, and clinical factors). These clarifications are now included in the revised manuscript.

Comment 7: Check assumptions of Cox regression more thoroughly. Although Schoenfeld residuals and VIFs are mentioned, results are not reported. Include at least a summary or figure confirming proportional hazard assumption and no multicollinearity.

Response 7: Thank you for this important observation. We have revised the Methods section to report the results of diagnostic checks. We now state that the proportional hazards assumption was tested using Schoenfeld residuals and showed no significant violations. We also report that all variance inflation factors (VIFs) in the fully adjusted model were <5, indicating no concerning multicollinearity.

Comment 8: Revise overly dense paragraphs. Some sections (e.g. lines 243–313) are overly dense and could be split for better readability. Use subheadings (e.g. Mechanisms, Prior Literature, Clinical Implications) in Discussion.

Response 8: Thank you for this helpful recommendation. In response, we revised the Discussion section to improve readability by breaking down overly dense paragraphs into shorter, thematically organized sections to guide the reader and enhance clarity. These changes are reflected in the revised manuscript.

Comment 9: Tighten grammar and phrasing

"Patients with PAD was also..." → “Patients with PAD were also...”

“This coexistence is associated with higher rates...” → “The coexistence is associated with...”

Watch subject–verb agreement and plural forms.

Response 9: Thank you for pointing this out. We carefully reviewed the manuscript for grammatical issues, including subject–verb agreement, plural forms, and awkward phrasing. Revisions were made throughout the text to improve clarity and accuracy. All changes have been incorporated into the revised manuscript.

Comment 10: Avoid vague phrases like “substantially heightened”. Use more precise quantitative language whenever possible, especially in scientific writing. Replace with actual HRs or % differences where appropriate.

Response 10: Thank you for this valuable suggestion. We have revised the manuscript to replace vague phrases such as “substantially heightened” with more precise quantitative language. Wherever possible, we now report specific hazard ratios (HRs), percentage differences, or effect estimates to clearly convey the magnitude of associations. These revisions have been incorporated throughout the Results and Discussion sections.

Comment 11: Improve clarity in mortality definitions. Distinguish clearly between "all-cause mortality" and "cardio-cerebrovascular mortality" at each mention, especially in the abstract, results, and conclusion.

Response 11: Thank you for this helpful comment. We have revised the manuscript to ensure consistent and clear distinction between all-cause mortality and cardio-cerebrovascular mortality throughout the abstract, results, and conclusion.

Comment 12: The discussion section repeats the same hazard ratios reported earlier. Instead, focus more on interpretation, implications, and mechanisms than reiterating the same numbers.

Response 12: Thank you for the valuable suggestion. We have revised the Discussion section to reduce redundancy and added brief statements highlighting the interpretation of our findings and their clinical implications, focusing on the importance of early identification and integrated management of patients with coexisting CKD and PAD.

Comment 13: Acknowledge potential residual confounding. Even after adjustment, acknowledge that unmeasured or residual confounding (e.g., medication use, healthcare access, inflammation markers) may influence results.

Response 13: Thank you for this important observation. As suggested, we have revised the Discussion to explicitly acknowledge the possibility of residual and unmeasured confounding, including factors such as medication use, healthcare access, and inflammatory markers, which were not available in our dataset. This limitation has been added to the Limitations subsection of the Discussion.

We hope these revisions meet the expectations of the reviewers and the editorial team. Thank you again for your time and consideration.

Sincerely,

Xuan Li, MD

Corresponding Author

Department of Vascular Surgery, Zhejiang Hospital, No. 12 Lingyin Road, Hangzhou, Zhejiang, 310013, China.

E-mail: lixuandr79@outlook.com

---

## [Decision Letter · Decision Letter 1]

3 Oct 2025

Dear Dr. li,

Thank you for submitting your manuscript to PLOS ONE. After careful consideration, we feel that it has merit but does not fully meet PLOS ONE’s publication criteria as it currently stands. Therefore, we invite you to submit a revised version of the manuscript that addresses the points raised during the review process.

We look forward to receiving your revised manuscript.

Kind regards,

Natural Hoi Sing Chu, Ph.D

Academic Editor

PLOS ONE

Journal Requirements:

Reviewers' comments:

Reviewer's Responses to Questions

**Comments to the Author**

Reviewer #1: All comments have been addressed

2. Is the manuscript technically sound, and do the data support the conclusions?

Reviewer #1: Yes

3. Has the statistical analysis been performed appropriately and rigorously?

Reviewer #1: Yes

4. Have the authors made all data underlying the findings in their manuscript fully available?

Reviewer #1: Yes

5. Is the manuscript presented in an intelligible fashion and written in standard English?

Reviewer #1: Yes

Reviewer #1: The authors have substantially improved the manuscript in response to prior reviewer comments. The use of a large, nationally representative NHANES dataset with long-term follow-up, combined with standardized definitions of CKD and PAD, strengthens the study’s rigor. The revisions have addressed many methodological, clarity, and reporting concerns, and the manuscript is now more concise, better organized, and clearer in its interpretation of findings. The work provides useful epidemiological evidence relevant to clinical risk stratification.

However, a few areas still require clarification or refinement before the manuscript is suitable for publication.

1.Novelty and Contribution

- The introduction now better highlights how this study extends prior work (dual CKD measures, ABI-defined PAD, long follow-up, and complex survey design).

- Suggestion: The authors may further emphasize the public health implications—i.e., how screening for both conditions together may identify a particularly high-risk population that merits more aggressive preventive strategies.

2. Statistical Analysis and Interpretation

- The inclusion of VIF values (<5) and reporting of Schoenfeld residual checks are appropriate.

- However, the authors should present the VIF results (perhaps in supplementary materials) rather than only stating them in text, for transparency.

- The conclusion now states the joint effect is “additive rather than synergistic,” which is more accurate. This is an important clarification, but the clinical framing (“double jeopardy”) still implies synergy—consider softening this phrasing further.

3. Mechanistic Discussion

- The mechanistic explanations are thorough, but still somewhat speculative. While the authors now clearly flag this as hypothesis-generating, they may consider shortening the mechanistic section to avoid overinterpretation and emphasize the epidemiological message of the findings.

4. Limitations

- The limitations are more complete, including unmeasured confounding (medication use, inflammatory markers, adherence).

- Suggestion: Add one sentence noting that ABI alone may misclassify PAD in populations with arterial stiffness (common in diabetes and CKD). This nuance would further strengthen the discussion.

5. Terminology

- Ensure consistent use of “cardio-cerebrovascular disease (CCD)” throughout the text. At times, “CVD” or “cardiovascular” appear instead, which may confuse readers.

6. References

- The reference list is up to date and comprehensive. However, some highly relevant systematic reviews on CKD–PAD overlap (e.g., meta-analyses of vascular risk in CKD populations) could be cited to further contextualize findings.

**Do you want your identity to be public for this peer review?** For information about this choice, including consent withdrawal, please see our Privacy Policy

Reviewer #1: No

---

## [Author Response · Author response to Decision Letter 2]

18 Oct 2025

Dear Editor and Reviewers,

We would like to express our sincere appreciation for your constructive and insightful comments on our manuscript. We have carefully considered each suggestion and revised the manuscript accordingly to improve its clarity and scientific rigor. In accordance with the journal’s submission requirements, we have submitted the revised manuscript in its final, clean format, without any highlighted or tracked changes. Below, we provide a detailed point-by-point response to each of the reviewers’ comments.

Reviewer #1:

Comment 1: Novelty and Contribution

- The introduction now better highlights how this study extends prior work (dual CKD measures, ABI-defined PAD, long follow-up, and complex survey design).

- Suggestion: The authors may further emphasize the public health implications—i.e., how screening for both conditions together may identify a particularly high-risk population that merits more aggressive preventive strategies.

Response 1: We appreciate your insightful suggestion. In response, we have revised the Introduction to further highlight the public health implications of our findings. We now emphasize how simultaneous screening for CKD and PAD using simple, widely available measures (eGFR, UACR, and ABI) may help identify a particularly high-risk subgroup that could benefit from targeted preventive interventions and multidisciplinary care.

Comment 2: Statistical Analysis and Interpretation

- The inclusion of VIF values (<5) and reporting of Schoenfeld residual checks are appropriate.

- However, the authors should present the VIF results (perhaps in supplementary materials) rather than only stating them in text, for transparency.

- The conclusion now states the joint effect is “additive rather than synergistic,” which is more accurate. This is an important clarification, but the clinical framing (“double jeopardy”) still implies synergy—consider softening this phrasing further.

Response 2: We sincerely thank the reviewer for these constructive comments. In accordance with the suggestions, we have added a new supplementary table (Supplementary Table S1) presenting the variance inflation factor (VIF) values for all covariates included in the multivariable Cox regression models. All VIF values were below 5, confirming the absence of significant multicollinearity and enhancing the transparency of our statistical analyses.

Additionally, we have revised the conclusion to soften the clinical phrasing and to emphasize the additive—rather than synergistic—nature of the observed joint effect. These revisions improve both the precision and clarity of the manuscript.

Comment 3: Mechanistic Discussion

- The mechanistic explanations are thorough, but still somewhat speculative. While the authors now clearly flag this as hypothesis-generating, they may consider shortening the mechanistic section to avoid overinterpretation and emphasize the epidemiological message of the findings.

Response 3: We thank the reviewer for this thoughtful recommendation. In response, we have shortened the mechanistic discussion to avoid overinterpretation and ensure alignment with the observational nature of our study. These revisions help improve the focus and clarity of the discussion.

Comment 4: Limitations

- The limitations are more complete, including unmeasured confounding (medication use, inflammatory markers, adherence).

- Suggestion: Add one sentence noting that ABI alone may misclassify PAD in populations with arterial stiffness (common in diabetes and CKD). This nuance would further strengthen the discussion.

Response 4: We thank the reviewer for this valuable suggestion. In response, we have revised the Limitations section to acknowledge that the use of ABI alone may lead to misclassification of PAD in individuals with arterial stiffness, particularly among those with diabetes or CKD. This addition provides a more nuanced interpretation and strengthens the discussion of study limitations.

Comment 5: Terminology

- Ensure consistent use of “cardio-cerebrovascular disease (CCD)” throughout the text. At times, “CVD” or “cardiovascular” appear instead, which may confuse readers.

Response 5: We thank the reviewer for this helpful observation. We have carefully reviewed the entire manuscript and revised the terminology to ensure consistent use of “cardio cerebrovascular disease (CCD)” throughout the text, tables, and figure legends to maintain clarity and coherence.

Comment 6: References

- The reference list is up to date and comprehensive. However, some highly relevant systematic reviews on CKD–PAD overlap (e.g., meta-analyses of vascular risk in CKD populations) could be cited to further contextualize findings.

Response 6: We appreciate the reviewer’s thoughtful suggestion. In response, we have incorporated additional citations to recent systematic reviews and meta‑analyses addressing the overlap between CKD and PAD and the associated vascular risk. These references have been added to the Introduction and Discussion section to provide broader context and strengthen the interpretation of our findings.

We hope these revisions meet the expectations of the reviewers and the editorial team. Thank you again for your time and consideration.

Sincerely,

Xuan Li, MD

Corresponding Author

Department of Vascular Surgery, Zhejiang Hospital, No. 12 Lingyin Road, Hangzhou, Zhejiang, 310013, China.

E-mail: lixuandr79@outlook.com

---

## [Editor Report · Decision Letter 2]

23 Oct 2025

The Combined Influence of Chronic Kidney Disease and Peripheral Artery Disease on Long-Term All-Cause and Cardio-cerebrovascular Disease Mortality Among Middle-aged and Elderly Individuals: A Nationwide Cohort Study

PONE-D-25-06158R2

Dear Dr. li,

We’re pleased to inform you that your manuscript has been judged scientifically suitable for publication and will be formally accepted for publication once it meets all outstanding technical requirements.

Kind regards,

Natural Hoi Sing Chu, Ph.D

Academic Editor

PLOS ONE
---

## [Editor Report · Acceptance letter]

PONE-D-25-06158R2

PLOS ONE

Dear Dr. li,

I'm pleased to inform you that your manuscript has been deemed suitable for publication in PLOS ONE. Congratulations! Your manuscript is now being handed over to our production team.

Kind regards,

on behalf of

Dr. Natural Hoi Sing Chu

Academic Editor

PLOS ONE